# Genome-wide DNA hypomethylation and RNA:DNA hybrid accumulation in Aicardi–Goutières syndrome

Yoong Wearn Lim[1], Lionel A Sanz[1], Xiaoqin Xu[1], Stella R Hartono[1], Frédéric Chédin[1,2]*

[1]Department of Molecular and Cellular Biology, University of California, Davis, Davis, United States; [2]Genome Center, University of California, Davis, Davis, United States

**Abstract** Aicardi–Goutières syndrome (AGS) is a severe childhood inflammatory disorder that shows clinical and genetic overlap with systemic lupus erythematosus (SLE). AGS is thought to arise from the accumulation of incompletely metabolized endogenous nucleic acid species owing to mutations in nucleic acid-degrading enzymes TREX1 (AGS1), RNase H2 (AGS2, 3 and 4), and SAMHD1 (AGS5). However, the identity and source of such immunogenic nucleic acid species remain undefined. Using genome-wide approaches, we show that fibroblasts from AGS patients with AGS1-5 mutations are burdened by excessive loads of RNA:DNA hybrids. Using MethylC-seq, we show that AGS fibroblasts display pronounced and global loss of DNA methylation and demonstrate that AGS-specific RNA:DNA hybrids often occur within DNA hypomethylated regions. Altogether, our data suggest that RNA:DNA hybrids may represent a common immunogenic form of nucleic acids in AGS and provide the first evidence of epigenetic perturbations in AGS, furthering the links between AGS and SLE.

*For correspondence: flchedin@ucdavis.edu

**Competing interests:** The authors declare that no competing interests exist.

**Reviewing editor**: Bing Ren, University of California, San Diego School of Medicine, United States

## Introduction

Aicardi–Goutières syndrome (AGS) is a severe inflammatory encephalopathy characterized by neurological dysfunction, psychomotor retardation, seizures, unexplained fevers, joint stiffness, basal ganglia calcification, and chilblain skin lesions (*Rice et al., 2007b*). Despite its rarity, AGS has captured the attention of the scientific community due to its clinical, molecular, and genetic overlap with another, common, systemic autoimmune disorder, systemic lupus erythematosus (SLE) (*Rice et al., 2007a, 2007b*; *Ravenscroft et al., 2011*) (*Gunther et al., 2015*). Both diseases are characterized, and to a large extent driven, by excessive expression of interferon alpha, a potent anti-viral cytokine associated with heightened innate immune response and inflammation (*Crow, 2011*). AGS is a monogenic disorder caused by mutations in any one of the six AGS genes. The AGS genes encode for the 3′ to 5′ single-stranded DNA exonuclease TREX1 (AGS1) (*Crow et al., 2006a*), the RNA:DNA hybrid-specific ribonuclease H2 subunits (RNase H2A, B and C, corresponding to AGS4, 2 and 3, respectively) (*Crow et al., 2006b*), the 3′ to 5′ exonuclease and dNTP hydrolase SAMHD1 (AGS5) (*Rice et al., 2009*; *Goldstone et al., 2011*), and the RNA adenosine deaminase ADAR1 (AGS6) (*Rice et al., 2012*). Recently, gain-of-function mutations in the cytosolic double-stranded RNA receptor gene IFIH1 have also been shown to be associated with AGS (*Rice et al., 2014*). Interestingly, mutations in *TREX1* and the *RNASEH2* genes have also been implicated in SLE, further illustrating the genetic link between AGS and SLE (*Lee-Kirsch et al., 2007*; *Gunther et al., 2015*). Given the role of AGS enzymes in DNA and RNA metabolism, the innate immune response in AGS is thought to be triggered by the accumulation of incompletely metabolized endogenous nucleic acid elements

**eLife digest** The immune system protects the body from attack by bacteria, viruses, and other microbes. A key feature of this system is the ability to discriminate between the body's own cells and potential foreign invaders. Occasionally, this process can go wrong and the immune system starts attacking its own tissues, which can lead to arthritis, diabetes, lupus, and other 'autoimmune' diseases.

Aicardi–Goutières syndrome (AGS) is an autoimmune disease that leads to severe mental and physical symptoms. Recent research has revealed that the disease is caused by mutations in genes that make enzymes called nucleases. In healthy people, these enzymes destroy DNA molecules and other nucleic acids. In AGS patients, the failure of the nucleases to act is thought to lead to the accumulation of unwanted DNA and RNA molecules. These molecules, in turn, are thought to be mistakenly identified by the immune system as 'foreign' and to cause an autoimmune response. However, it is not clear how this works.

Here, Lim et al. studied skin cells called fibroblasts from patients with Aicardi–Goutières syndrome. The experiments found that the patients' cells had excessive numbers of RNA molecules binding to sections of matching DNA. These unusual DNA–RNA 'hybrids' accumulated in regions of the genome that do not contain many genes, perhaps as a result of breaks in the DNA. It is possible that they may mimic nucleic acids from viruses and could trigger an autoimmune response.

In healthy individuals, small 'methyl' groups are often attached to DNA in a process known as DNA methylation. This serves to maintain the stability of the genome and controls the activity of genes. Unexpectedly, Lim et al. found that the DNA in AGS patients had far fewer methyl groups, especially in areas where the DNA–RNA hybrids had accumulated. This may lead to genome destabilization, alterations in gene activity, and may mean that the DNA in these regions may be mistaken for foreign DNA by the immune system.

Altogether, Lim et al.'s findings suggest that Aicardi–Goutières syndrome may be caused by immune responses triggered by the accumulation of RNA–DNA hybrids and lower levels of DNA methylation. These findings may aid the development of new therapies to treat Aicardi–Goutières syndrome, lupus, and other similar diseases.

(*Crow and Rehwinkel, 2009*). However, the identity and source of such immunogenic nucleic acid elements remain undefined.

While genetically well defined, the potential contribution of epigenetic deregulation to AGS has not been addressed. Characterizing putative epigenetic deregulation in AGS is all the more justified given that DNA hypomethylation, either spontaneous or caused by drug exposure, triggers increased B cell autoreactivity and the development of drug-induced and idiopathic human lupus (*Richardson, 1986*; *Quddus et al., 1993*; *Yung et al., 1995*; *Jeffries et al., 2011*; *Absher et al., 2013*; *Zhang et al., 2013*). Furthermore, loss of DNA methylation, albeit modest and restricted to specific autoimmunity-related genes, was reported in SLE patients (*Richardson et al., 1990*; *Ballestar et al., 2006*; *Javierre et al., 2010*; *Jeffries et al., 2011*; *Absher et al., 2013*), raising the question of whether the same epigenetic deregulation is shared in AGS.

To uncover the mechanism driving inflammatory responses in AGS, we sought to identify transcriptional, genetic, and epigenetic perturbations shared among AGS subtypes. For this, we extensively profiled a series of primary fibroblast cells from AGS patients with mutations in AGS1, AGS2, and AGS4 and AGS5 using high-throughput sequencing technologies. Transcriptional profiling via RNA-seq showed that AGS fibroblasts are characterized by an activated antiviral, immune response with a characteristic interferon signature. Genome-wide profiling of RNA:DNA hybrid formation revealed that AGS cells accumulate excessive loads of RNA:DNA hybrids. Whole-genome DNA methylation profiling further indicated that AGS genomes experience global loss of DNA methylation. Our data therefore provide the first evidence that *TREX1*, *RNASEH2*, and *SAMHD1* mutations share common molecular abnormalities including genome-wide DNA hypomethylation and accumulation of RNA:DNA hybrid species.

## Results

### Primary AGS fibroblasts show heightened immunomodulatory transcriptional responses

We first performed RNA-seq to ascertain transcriptional signatures of AGS using primary fibroblasts from four patients with mutations in AGS1, AGS2, AGS4, or AGS5, and an age-matched healthy control (see Supplementary file for detailed genotype information). We identified a total of 98 and 209 genes that were significantly up- and down-regulated, respectively, in at least one AGS sample (*Figure 1A*; *Figure 1—figure supplement 1A*; *Supplementary file 2*). AGS down-regulated genes significantly associated with ontologies linked to cell adhesion and the extracellular matrix (*Supplementary file 2*). These ontologies may be relevant to the skin lesions often observed in AGS and SLE patients (*Rice et al., 2007b*). Genes involved in inflammation, immune responses, chemokine signaling pathways, and sensing of viral nucleic acids were up-regulated in AGS patient cells. Genes for the major pro-inflammatory cytokine IL1β (*Zhao et al., 2013b*) and the CXCL5 and CXCL6 chemokines were up-regulated in several AGS patients (*Figure 1A,B*). Additional immune signaling genes (*IL-33, CXCL3, CXCL1, IL-8, CCL11*) were significantly up-regulated in at least one AGS sample (*Supplementary file 2*) and gene ontology analysis of AGS deregulated genes identified cytokine and chemokine signaling pathways as one of weakly enriched terms (*Figure 1—figure supplement 1B*). These observations are indicative of a heightened immunomodulatory and chemotactic response previously identified in multiple autoimmune conditions, including SLE (*Tuller et al., 2013*). Genes involved in anti-viral responses (*RSAD2, OASL, IGF2BP1, BST2*), in particular interferon-inducible genes, were also up-regulated in multiple AGS samples (*Figure 1A,B*). Additional interferon-inducible genes (*IFI6, IF44L, ISG15*) described previously as representing an interferon signature in SLE (*Baechler et al., 2003*) and AGS (*Rice et al., 2013*) showed up-regulation in at least one AGS sample (*Supplementary file 2*). AGS fibroblasts thus exhibit an activated immune, inflammatory, and anti-viral state, underscoring the systemic nature of the disease and indicating that fibroblasts are an appropriate cell type to study AGS.

### Accumulation of ribonucleotides in genomic DNA is unique to RNASEH2-deficient AGS patients

While it is likely that dysfunction in AGS enzymes results in an accumulation of incompletely metabolized immunogenic nucleic acid species (*Crow and Rehwinkel, 2009*), the identity and source of these molecules remain unclear. RNase H2 generally degrades RNA:DNA hybrids (*Cerritelli and Crouch, 2009*) and is specifically responsible for removing single ribonucleotides that are misincorporated during DNA replication (*Reijns et al., 2012*). We therefore determined whether ribonucleotide accumulation is a common feature of AGS. For this, we treated genomic DNA from AGS and control primary fibroblasts with purified human RNase H2 and measured the presence of resulting nicks by DNA polymerase I-dependent nick translation in the presence of [α-$^{32}$P] dCTP. Genomic DNA from wild-type and RNase H2A-deficient (*rnh201Δ*) yeast cells that readily incorporate ribonucleotides (*Nick McElhinny et al., 2010*) was used as a control. Relative to wild-type, *rnh201Δ* yeast cells showed a fourfold to fivefold increase in ribonucleotide accumulation (*Figure 2A,B*), consistent with prior results (*Nick McElhinny et al., 2010*). Cells from two patients mutated in *RNASEH2B (AGS2)* showed a twofold to threefold increase in labeling (*Figure 2A,C*). Cells from two patients mutated in the catalytic RNase H2A subunit *(AGS4)* showed a pronounced 10–25-fold increase in ribonucleotides (*Figure 2A,C*). Thus, as observed in yeast (*Nick McElhinny et al., 2010*) and mouse (*Reijns et al., 2012*) models, a reduction of human RNase H2 activity leads to elevated levels of ribonucleotides in genomic DNA. By contrast, no significant increase in ribonucleotide loads could be detected in patients mutated in *TREX1 (AGS1)* or *SAMHD1 (AGS5)* (*Figure 2A,B*). Thus, while aberrant accumulation of ribonucleotides may contribute to the severity of the disease in RNase H2-defective AGS patients, it is unlikely to represent a common form of disease-causing nucleic acids in AGS.

### RNA:DNA hybrids accumulate in AGS patients

To identify other species of RNA:DNA hybrids that may be accumulating in AGS, we performed DRIP-seq, a technique originally developed to profile R-loop formation genome-wide (*Ginno et al., 2012*). R-loops are long RNA:DNA hybrid structures that form co-transcriptionally upon re-annealing of the RNA

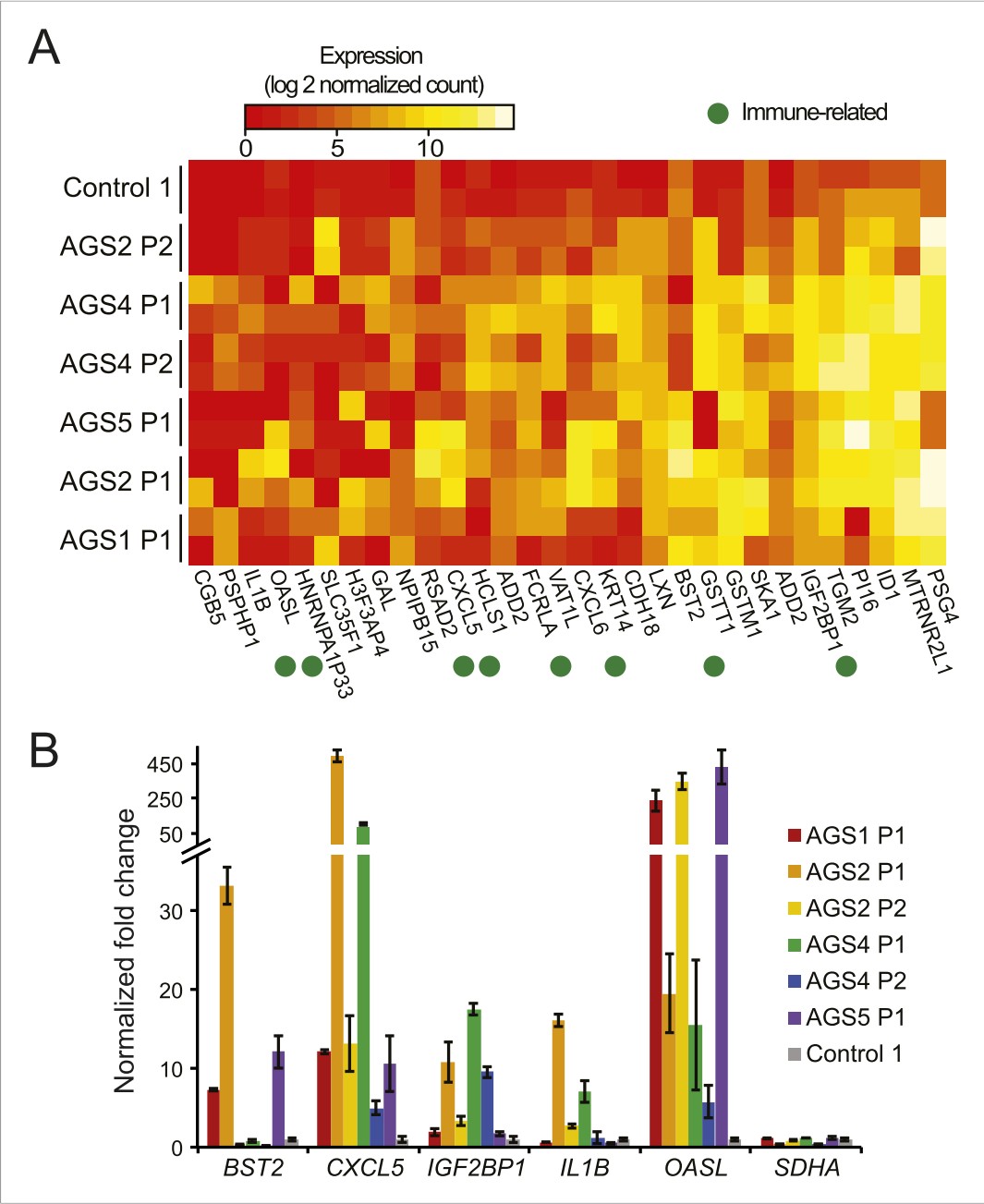

**Figure 1**. AGS fibroblasts exhibit activated immune antiviral response. (**A**) RNA-seq heatmap showing log2 expression level of genes up-regulated in at least two Aicardi–Goutières syndrome (AGS) samples. Two biological replicates are shown for each sample. Immune-related genes are marked with green dots. (**B**) Gene expression changes between AGS samples and control were measured for five immune-related genes by real time reverse-transcription PCR (RT-qPCR). Y-axis represents fold change in gene expression, normalized to *GAPDH* gene, relative to the control sample. *SDHA* is another housekeeping gene used as control. Each value is the average of at least two technical replicates. Error bars represent SEM.

The following figure supplement is available for figure 1:

**Figure supplement 1**. RNA-seq of AGS and control fibroblasts.

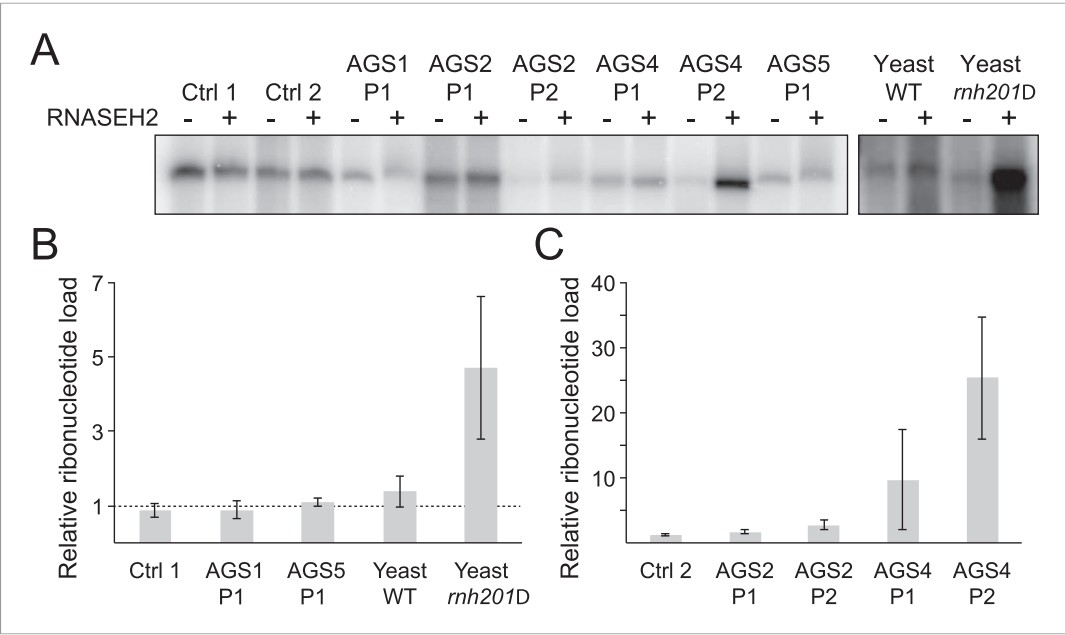

**Figure 2**. Incorporation of ribonucleotides in genomic DNA is only observed in RNase H2-deficient AGS fibroblasts. (**A**) Genomic DNA was treated with RNase H2 to reveal the presence of ribonucleotides as single-strand nicks. An untreated control was used to reveal background single-stranded breaks in genomic DNA preparations. The intensity of labeling after gel electrophoresis was measured. (**B**, **C**) Relative ribonucleotide loads are reported as fold-increase of radiolabel incorporation of RNase H2-treated over untreated sample. At least three independent replicates were performed and the error bars indicate SEM.

transcript to the template DNA strand, forcing the non-template strand into a single-stranded state. R-loops are enriched at the 5′- and 3′-ends of human genes where they play roles in epigenetic control and transcription regulation (*Ginno et al., 2012*, *2013*). DRIP-seq revealed that AGS and control fibroblasts share a total of 15,897 DRIP peaks representing approximately 141 megabases of genomic space (*Figure 3A,C*, *Figure 3—figure supplement 1A,B*). Consistent with prior studies, 32% of these peaks mapped to promoters or gene ends, which represents a strong enrichment over expected genomic distribution and further indicates that gene ends correspond to R-loop formation hotspots (*Ginno et al., 2013*). Our high-coverage DRIP-seq data also revealed that 56% of the R-loop signal mapped onto gene bodies, suggesting that RNA–DNA entanglements during transcription are prevalent (*Figure 3—figure supplement 1C*). By contrast, only 15.6% of common R-loop peaks mapped to intergenic regions, consistent with a predominant genic origin for R-loop formation (*Figure 3—figure supplement 1C*). Overall, common DRIP peaks showed strong overlap with GC skew (*Figure 3B*), a key sequence determinant that favors co-transcriptional R-loop formation (*Ginno et al., 2013*). The observation that AGS samples display R-loop formation over expected R-loop forming, GC-skewed, regions indicates that canonical R-loop formation is not significantly altered in AGS patients (*Figure 3—figure supplement 1A,B*).

Despite the relative agreement between AGS and control R-loop profiles over a broad set of common peaks, a large excess of DRIP peaks (33,781 in total) was observed specifically in AGS cells (*Figure 3A*). These novel peaks of RNA:DNA hybrids were in many cases unique to each patient sub-type. While a small number of DRIP peaks were also unique to the control sample, AGS-specific DRIP peaks in each AGS subtype were both threefold to fourfold greater in numbers, and when combined, occupied a three to four times larger genomic space than control-specific peaks (*Figure 3C*). On average, AGS-specific DRIP peaks occupied an additional 23.8 megabases of DNA sequence. This indicates that all AGS subtypes are burdened by higher loads of RNA:DNA hybrids in their genomes.

Unlike common DRIP peaks, AGS-specific DRIP peaks were threefold to sevenfold less likely to overlap with GC skew (*Figure 3B*), suggesting that they may originate according to a non-canonical mechanism. Additionally, AGS-specific DRIP peaks were depleted at transcription start sites (TSSs)

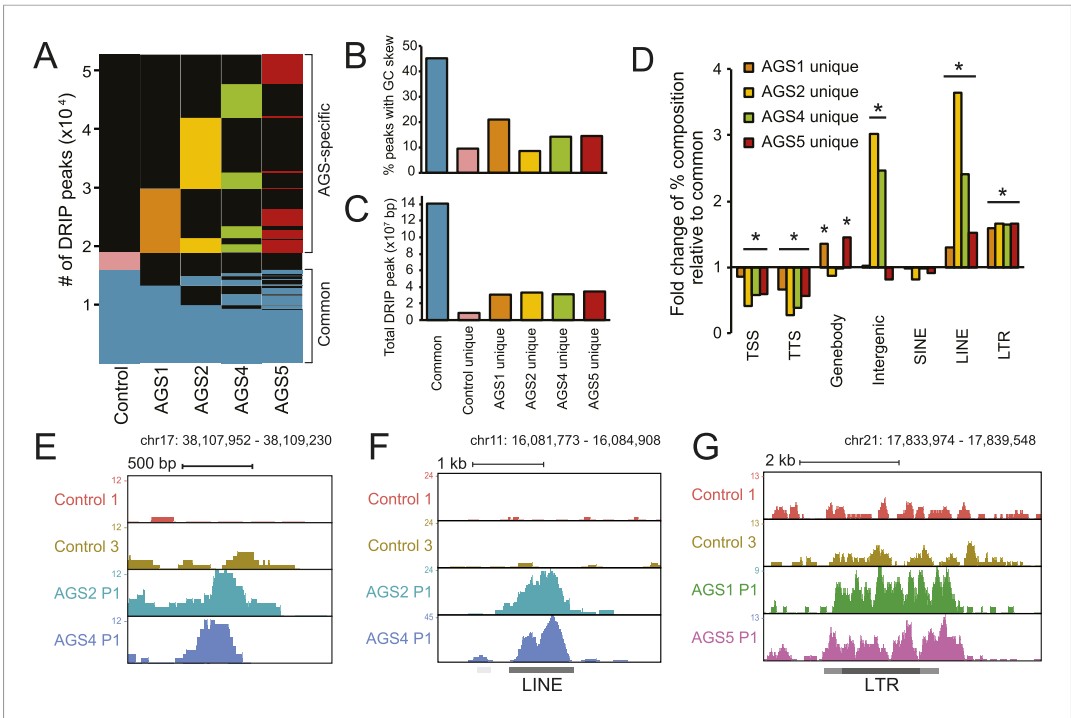

**Figure 3**. AGS fibroblasts accumulate RNA:DNA hybrids. (**A**) All genomic loci overlapping with a DRIP peak in at least one sample are stacked vertically; the position of each peak in a stack is constant horizontally across samples. Each patient subtype or control occupies a vertical bar, as labeled. Each bar corresponds to merged data sets from two independent samples. Common peaks (i.e., form in control and at least one AGS sample) are represented in blue. Control-unique DRIP peaks are shown in pink; lack of DRIP signal over a given peak in any sample is shown as black. AGS-unique peaks are colored orange, yellow, green, and red in AGS1, 2, 4, and 5, respectively. Brackets on the right side demarcate common and AGS-specific peaks, respectively. (**B**, **C**) Graphs showing the % overlap between DRIP peaks and blocks of GC skew (**B**); and the total size of DRIP peaks in each category (**C**). Color codes are as described for (**A**). (**D**) Enrichment or depletion of AGS-unique DRIP peaks over different genomic features is shown relative to common DRIP peaks. * indicates p < 0.002 and fold change >20% relative to common peaks. (**E**–**G**) Representative examples of AGS-specific DRIP peaks over an intergenic region (**E**), a truncated long interspersed nuclear elements (LINE) element (**F**) and a truncated long terminal repeats (LTR) element (**G**).

The following figure supplement is available for figure 3:

**Figure supplement 1**. Canonical R-loop genomic patterns are not affected in AGS fibroblasts.

and transcription termination sites (TTSs) (*Figure 3D*), where R-loops are typically observed (*Ginno et al., 2013*). In contrast, AGS2- and AGS4-specific peaks were significantly enriched over intergenic portions of the human genome (*Figure 3D,E*). AGS1- and AGS5-specific DRIP peaks were instead enriched over gene body regions (*Figure 3D*, *Figure 3—figure supplement 1D*). Interestingly, all AGS-specific DRIP peaks were significantly enriched in repeat classes corresponding to long interspersed nuclear elements (LINE) and long terminal repeats (LTR) retrotransposons (*Figure 3D,F,G*, *Figure 3—figure supplement 1D*). Analysis of the overlap of AGS-specific DRIP peaks with human-specific, retrotransposition-competent, LINE-1 elements failed to reveal any significant trend (data not shown). Therefore, whether the reported enrichment of AGS-specific DRIP peaks over LINE and LTR repeats carries biological significance or is simply a reflection of the increased repeat content of intergenic and intronic space remains to be determined. Altogether, our results indicate that AGS mutations in *TREX1*, *RNASEH2A*, *RNASEH2B*, and *SAMHD1* are associated with the accumulation of RNA:DNA hybrids over repeat-rich intergenic and gene body regions.

## AGS patients show pronounced genome-wide DNA hypomethylation

The observation that RNA:DNA hybrids accumulate over intergenic regions is surprising given that they are normally maintained in a transcriptionally quiescent state owing to the deposition of silencing

epigenetic marks such as DNA methylation (*Yoder et al., 1997*). To determine if DNA methylation patterns were altered in AGS, we performed MethylC-seq (*Lister et al., 2009*) to profile DNA methylation at low coverage (3.21–7.82 X coverage) across the same set of primary fibroblasts studied above (*Supplementary file 1*). All AGS cells displayed significant, global, DNA hypomethylation (*Figure 4A*). AGS2 and 4 cells showed profound DNA hypomethylation, with a ~20% reduction in

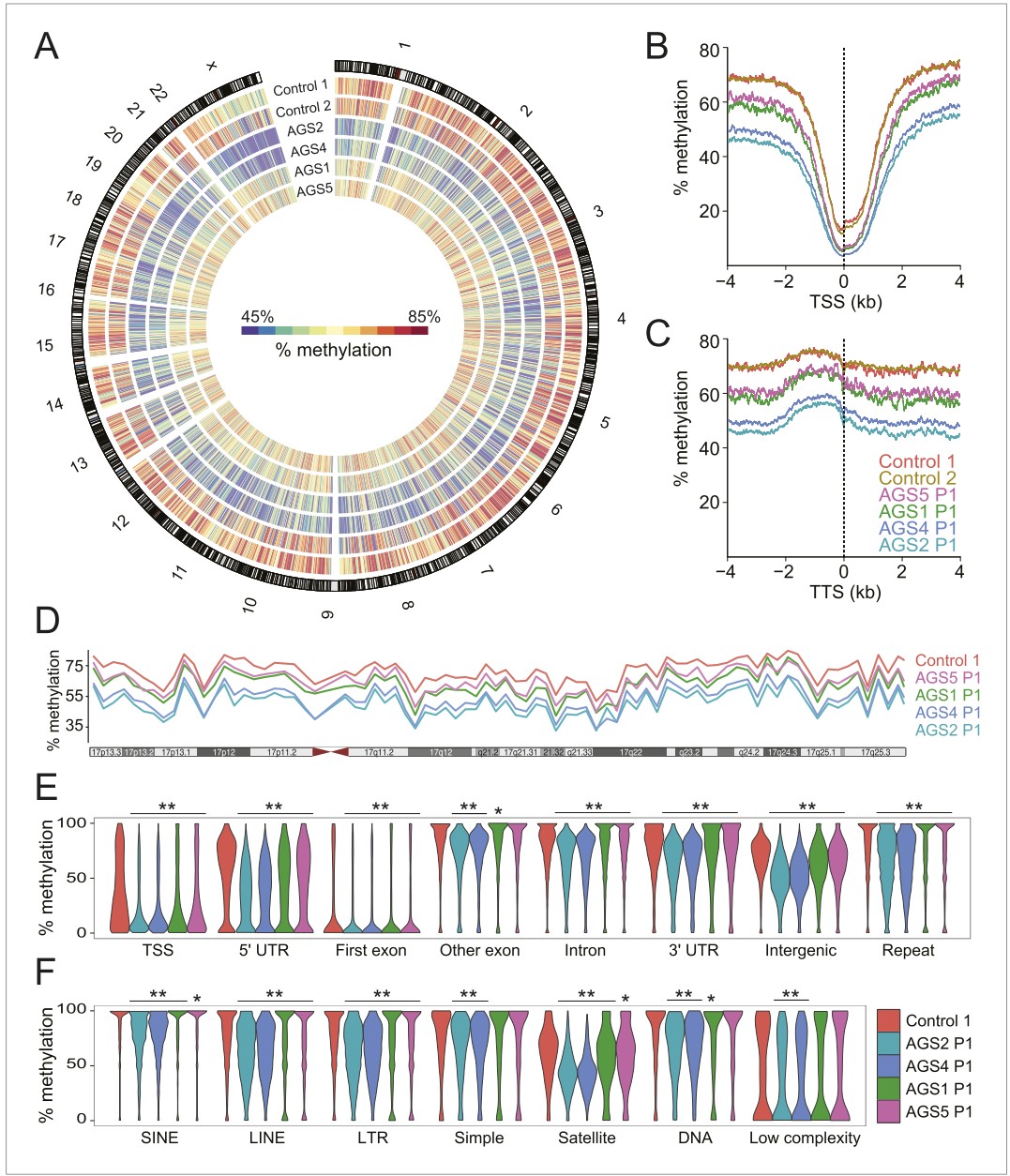

**Figure 4**. Genome-wide DNA hypomethylation in AGS fibroblasts. (**A**) Circos plot depicting DNA methylation along human chromosomes. Each tick mark is colored according to the average percent methylation across a 2 Mb genomic region (see color legend). (**B**, **C**) Metaplots of DNA methylation around transcription start site (TSS) (**B**) and transcription termination site (TTS) (**C**). (**D**) Percent methylation along chromosome 17. (**E**, **F**) Violin plots depicting methylation levels at different genomic (**E**) and repeat (**F**) regions. p-values are relative to control. **p < 1e-9, *p < 0.01.

The following figure supplement is available for figure 4:

**Figure supplement 1**. Methylation profiles of AGS and control cells.

genomic methylation levels overall. AGS1 and 5 showed a more moderate, but still highly significant, 5–10% reduction in DNA methylation. This reduction affected TSSs and TTSs (*Figure 4B,C*; *Figure 4—figure supplement 1A*) and spread along the lengths of entire chromosomes (*Figure 4D*), significantly impacting nearly all genomic compartments, including genic, intergenic, and repeat regions (*Figure 4E*). Analysis of repeat classes further revealed that LINE and LTR elements were significantly hypomethylated in all AGS subtypes (*Figure 4F*). Pyrosequencing assays targeting human-specific LINE-1 elements also revealed a small but significant decrease in DNA methylation over 4 CpG sites carried at the LINE-1 5′-UTR promoter (*Figure 4—figure supplement 1B*). SINE elements, satellite repeats, and other repeat types were significantly hypomethylated in AGS2 and 4 but not always in AGS1 and 5 (*Figure 4F*). This profound reduction in DNA methylation was not caused by any detectable deregulation in the expression of genes encoding for the DNA methylation machinery or of components of a putative DNA de-methylation system (*Figure 4—figure supplement 1C*).

To further characterize the DNA hypomethylation observed in AGS cells, we employed a Hidden Markov Model (HMM) to annotate highly and partially methylated domains (HMDs and PMDs, respectively) (*Schroeder et al., 2011*). PMDs in control fibroblasts were generally shared by AGS fibroblasts but not by embryonic stem (ES) cells (*Figure 5A*; *Figure 5—figure supplement 1A*), consistent with the notion that they represent fibroblast-specific PMDs. AGS2 and AGS4 fibroblasts displayed broader PMDs than the control cells, while the size of PMD regions in AGS1 and AGS5 cells was relatively unchanged (*Figure 5A*; *Figure 5—figure supplement 1A*). Analysis of the chromatin features associated with PMDs showed that common PMDs resided within silenced genomic regions enriched for Lamin-B1, a component of the nuclear lamina, and the silencing mark histone H3 Lysine 9 trimethylation (H3K9me3), as described previously (*Berman et al., 2012*). Common PMDs were also strongly depleted for the H3K27me3 mark, a modification associated with facultative heterochromatin and silencing over developmental genes (*Figure 5B*, *Figure 5—figure supplement 1B*). Interestingly, AGS-specific PMDs, while still associated with Lamin-B1 and to some extent with H3K9me3, were enriched for H3K27me3, particularly over regions corresponding normally to large H3K27me3 blocks (*Figure 5B,C*). This suggests that the DNA hypomethylation observed in AGS is particularly associated with regions of silenced, condensed chromatin.

We next determined if the regions that became hypomethylated also corresponded to regions that accumulated RNA:DNA hybrids in DRIP-seq analysis. In AGS1 and AGS5 fibroblasts, which showed only moderate DNA hypomethylation and limited additional PMDs, DRIP peaks, whether common or AGS-unique, mostly landed in HMD regions (*Figure 5D*). This is consistent with AGS1- and AGS5-specific DRIP peaks being enriched over gene body regions (*Figure 3D*) since gene body regions are typically highly methylated (*Lister et al., 2009*). In sharp contrast, nearly half of the AGS-unique DRIP peaks in AGS2 and AGS4 fibroblasts landed on PMD regions while only 10–15% of common DRIP peaks matched to PMDs in these two samples. Furthermore, two-thirds of these PMD-contained AGS-unique DRIP peaks matched to AGS-specific PMDs (*Figure 5D,E*). Consistent with this, the DNA methylation levels measured over AGS1-, AGS2-, and AGS4- (but not AGS5) unique DRIP peaks were significantly lower in their own respective samples compared to the DNA methylation levels of that same regions measured in control (*Figure 5F*). Thus, our data show a remarkable agreement between regions that accumulate RNA:DNA hybrids in the AGS2, AGS4, and to a lesser extent AGS1 subtypes and regions that undergo DNA hypomethylation.

## RNASEH2 depletion triggers DNA hypomethylation

To determine if the loss of DNA methylation observed in AGS patient cells could be directly due to AGS mutations, we focused on *RNASEH2A* since AGS4 fibroblasts displayed profound DNA hypomethylation (*Figure 4*). We employed a lentivirus-mediated CRISPR/Cas method to knockout exon 6 in *RNASEH2A* (RNASEH2A KO) in a HeLa cell line harboring a silent green fluorescent protein (GFP) reporter gene (HeLa-GFP) (*Poleshko et al., 2010*). As a control, we treated the same cell line with a CRISPR vector containing a scramble guide sequence (scramble). Effective gene knockout was validated by PCR genotyping (not shown) and by Western blots (*Figure 6A*). To further validate the knockout, we measured the DNA damage response in RNASEH2A KO and scramble cells using an antibody against gamma H2AX. As expected from previous reports (*Reijns et al., 2012*; *Gunther et al., 2015*), we observed higher levels of gamma H2AX in RNASEH2A KO cells (*Figure 6B*), indicative of an activated DNA damage signaling.

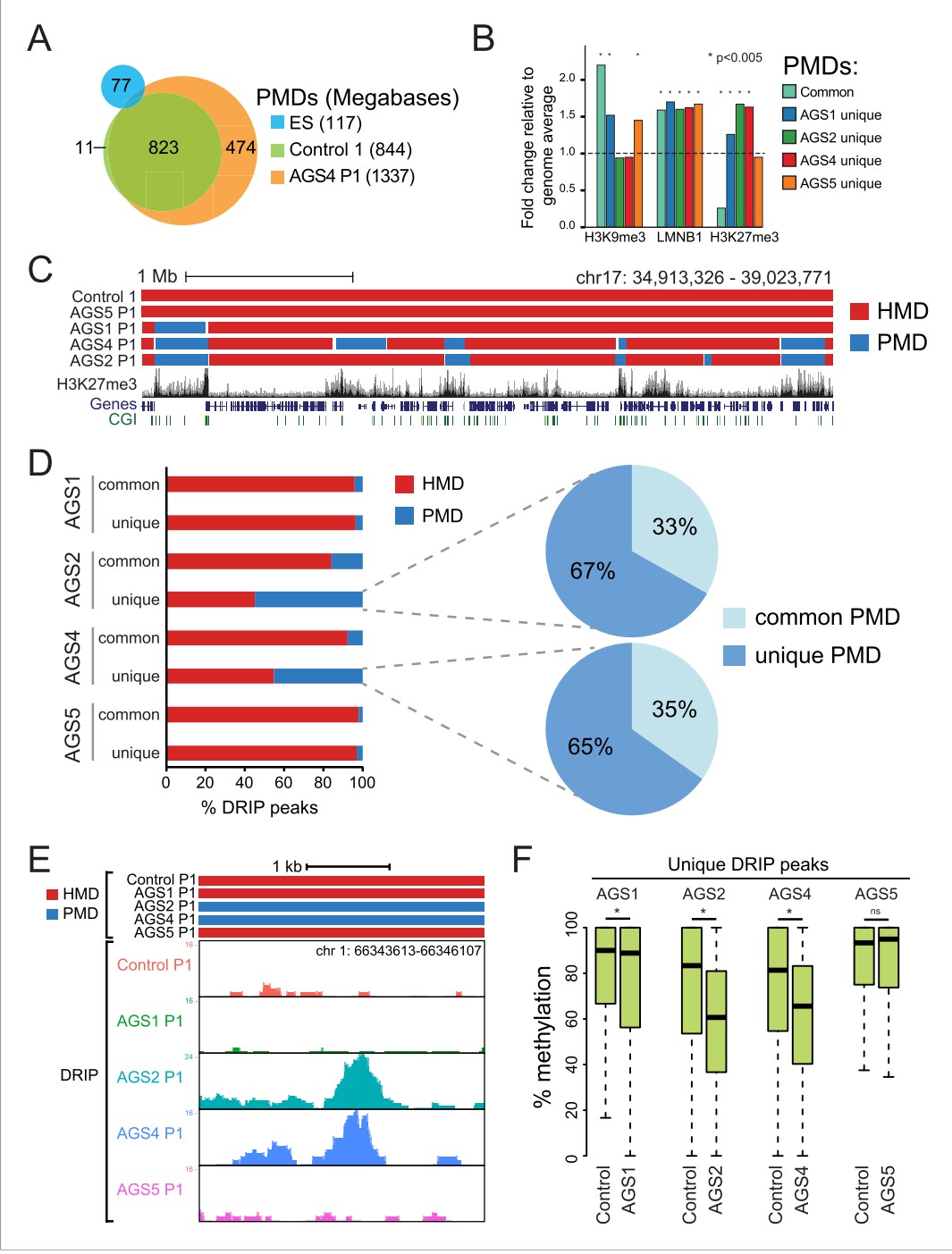

**Figure 5**. AGS-specific PMDs are enriched over H3K27me3-marked genomic regions. (**A**) Venn diagram displaying partially methylated domains (PMDs) in AGS4 and control fibroblasts compared to embryonic stem (ES) cells. (**B**) Location analysis of PMDs over regions marked by H3K9me3, Lamin-B1, and H3K27me3, shown as fold change relative to genome average (represented by the dotted horizontal line). Stars indicate the p-value of the deviation from the genome average. (**C**) Genome browser screenshot showing overlap of AGS-specific PMDs with H3K27me3. (**D**) Percent DRIP peaks (common or AGS-unique) in highly methylated domains (HMDs) and PMDs. The pie charts break down AGS2/4 PMDs to common and unique PMDs. (**E**) Genome browser screenshot showing overlap between AGS2/4 PMDs with AGS2/4 DRIP-seq peaks. (**F**) Percent DNA methylation of AGS-unique DRIP peaks measured in control and AGS fibroblasts (as indicated at the bottom of the graph). *p < 7.18e-11, ns = not significant.

*Figure 5. continued on next page*

*Figure 5. Continued*

The following figure supplement is available for figure 5:

**Figure supplement 1**. HMDs and PMDs in AGS fibroblasts.

Compared to scramble, RNASEH2A KO cells showed a clear increase in the number and brightness of GFP-positive cells (*Figure 6C*). GFP expression in RNASEH2A KO cells was 14-fold higher than in scramble cells, as measured by Western blot (*Figure 6D*), demonstrating reactivation of the GFP reporter gene. To survey DNA methylation levels in RNASEH2A KO and scramble cells, we focused on two 5′-UTR portions common to LINE-1 elements and performed DNA methylation sequencing after sodium bisulfite treatment. RNASEH2A KO cells showed significant DNA hypomethylation relative to scramble (*Figure 6E,F*). This suggests a direct effect of RNASEH2A deficiency in triggering DNA hypomethylation and gene reactivation at least for the loci studied here.

## Discussion

One of the key observations from this study is that primary fibroblasts from all four AGS subtypes studied here experienced global loss of DNA methylation. While additional samples encompassing other AGS mutations subtypes will need to be studied to confirm this trend, this observation nonetheless reinforces the similarities between AGS and SLE, although the DNA hypomethylation reported here is stronger than that previously reported in SLE. In AGS, DNA hypomethylation was found to affect every genomic compartment, including genic and intergenic regions, unique and repeated sequences (*Figure 4*). In AGS2 and AGS4, where the effect was most marked, the DNA hypomethylation led to the formation of AGS-specific PMDs in addition to those normally observed in fibroblasts (*Lister et al., 2009*). Similar to previously characterized PMDs in somatic tissues and cancer samples, common fibroblasts PMDs described here overlapped with inactive, late-replicating, heterochromatic regions associated with H3K9me3 and Lamin-B1 (*Hawkins et al., 2010*; *Berman et al., 2012*). Interestingly, AGS-specific PMDs, but not common PMDs, were also enriched over regions marked by H3K27me3, a mark of facultative heterochromatin associated with developmental silencing (*Figure 5*). This suggests that silent, condensed genomic regions, may in fact be intrinsically prone to becoming PMDs, and that this tendency is made worse by AGS2 and AGS4 mutations. Our data further indicate that RNase H2 defects directly drive these epigenetic perturbations given that we could recapitulate DNA hypomethylation at a subset of LINE-1 sequences and trigger the reactivation of a silent GFP reporter by knocking out *RNASEH2A* (*Figure 6*). While the precise mechanism responsible for DNA hypomethylation in AGS remains to be elucidated, it is worth noting that defects in TREX1, RNase H2, and SAMHD1 all trigger the DNA damage response and are associated with increased genomic instability (*Yang et al., 2007*; *Reijns et al., 2012*; *Clifford et al., 2014*; *Kretschmer et al., 2015*). RNase H2 in fact associates with the DNA replication fork through the PCNA (proliferating cell nuclear antigen) clamp (*Bubeck et al., 2011*) and TREX1 was found to translocate to the nucleus upon replication stress where it might be involved in processing aberrant DNA replication intermediates (*Yang et al., 2007*). It is therefore possible that the replication stress caused by AGS mutations indirectly results in defects in replication-coupled, DNMT1-mediated DNA methylation maintenance (*Figure 7*) (*Law and Jacobsen, 2010*). In support of this notion, a recent report shows that the PCNA clamp unloads from the lagging strand of stalled DNA replication forks (*Yu et al., 2014*). Given that the DNMT1 maintenance DNA methyltransferase binds to PCNA (*Leonhardt et al., 1992*) and requires PCNA binding for efficient activity (*Schermelleh et al., 2007*), it is plausible that replication stress may lead to inefficient DNA methylation maintenance, particularly in silent, compact regions of the genome such as heterochromatic regions. An analogous mechanism was proposed to account for how defective DNA replication may trigger epigenetic instability by uncoupling DNA synthesis from histone recycling (*Sarkies et al., 2010*). Interestingly, 5-aza-2′-deoxycytidine, a drug best known as a DNA-demethylating agent, in fact causes genome-wide DNA damage (*Karpf et al., 1999*; *Palii et al., 2008*; *Orta et al., 2013*). Finally, pronounced DNA hypomethylation is also known to cause genomic instability (*Jackson-Grusby et al., 2001*; *Chen et al., 2007*; *Liao et al., 2015*) suggesting a possible positive feedback loop that may link and amplify defects in either pathways (*Figure 7*).

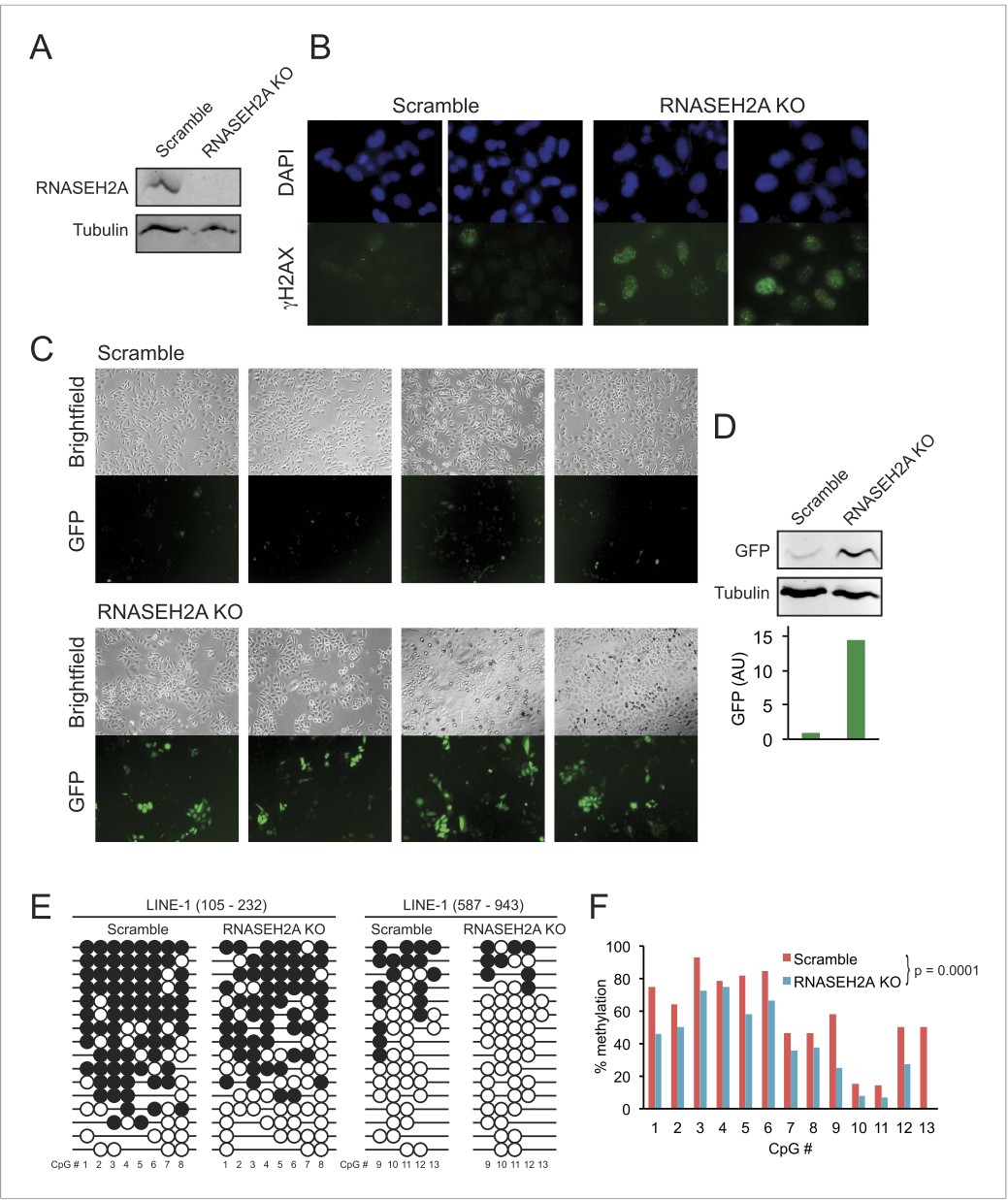

**Figure 6**. CRISPR/Cas-mediated RNASEH2A knockout induces DNA hypomethylation. (**A**) Western blot using an anti-RNase H2A antibody shows successful gene knockout in HeLa-GFP cells. Scramble cells were generated with a CRISPR vector carrying a scrambled guide RNA sequence. Tubulin was used as a loading control. (**B**) Immunocytochemistry confirms elevated DNA damage response in RNASEH2A KO cells; DAPI (blue), gamma H2AX antibody (green). (**C**) Bright field and green fluorescent protein (GFP) microscopy images of RNASEH2A KO and scramble control cells. Knockout of the *RNASEH2A* gene triggers the reactivation of the silent GFP reporter in HeLa-GFP cells. (**D**) (Top) Western blot showing GFP expression in HeLa-GFP cells (RNASEH2A KO and scramble); (bottom) bar graph quantification of Western blot. (**E**) Bisulfite methylation sequencing for two different LINE-1 loci in RNASEH2A KO and scramble control cells. Black and white circles represent methylated and unmethylated CpG sites, respectively. Missing bubbles indicate that a CpG site is absent from the sequence of that particular molecule. The coordinates for each fragment analyzed are indicated at top. (**F**) Quantification of percent methylation at each CpG sites surveyed in panel **E**. p-value was calculated using a paired Wilcoxon test with the alternative hypothesis that RNASEH2A KO is less methylated than scramble.

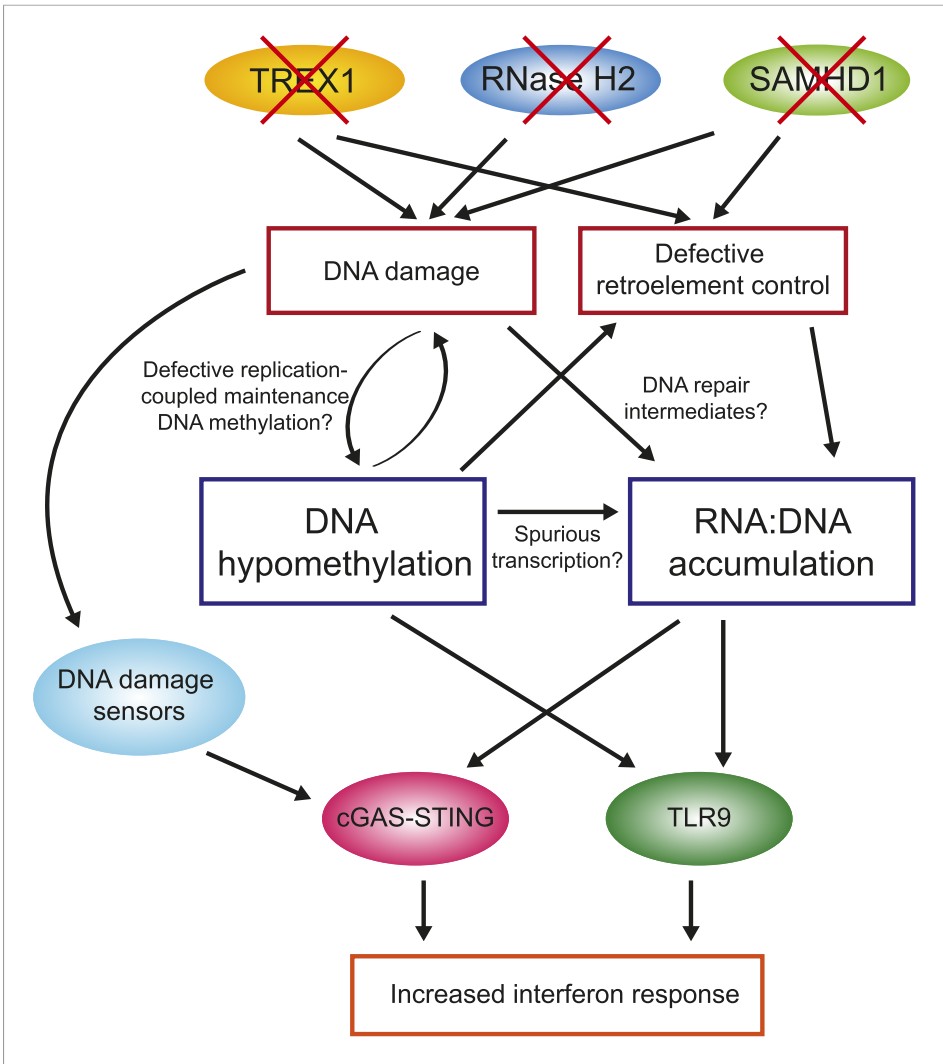

**Figure 7**. Overall model by which AGS1-5 mutations may lead to DNA hypomethylation, RNA:DNA hybrid accumulation, and IFN-stimulated immune response. See 'Discussion' for details.

It will be interesting to further investigate the connections between DNA replication stress, DNA hypomethylation, and epigenetic instability in the context of AGS or of other diseases associated with epigenetic aberrations such as cancers.

Understanding if and how DNA hypomethylation contributes to autoimmunity in AGS and SLE is more than ever a priority. One possibility is that DNA hypomethylation results in altered gene expression profiles. However, our RNA-seq characterization only identified a modest number of up- and down-regulated genes that were enriched for ontologies reflecting the biology of the disease itself, such as immunomodulatory genes (*Figure 1*). There is therefore no evidence for a global alteration of transcriptional profiles as a result of the DNA hypomethylation in AGS. This is consistent with the general absence of direct connections between DNA hypomethylation and gene reactivation (*Bestor et al., 2015*), including in extremely hypomethylated mouse ES cells (*Li et al., 1992*; *Tsumura et al., 2006*). It remains possible, however, that DNA hypomethylation may enable the transcription of normally silent and methylated retrotransposable elements. DNA methylation is indeed the primary silencing force ensuring retrotransposon control in somatic cells (*Walsh et al., 1998*; *Yoder and Bestor, 1998*). If correct, this could lead to the reactivation of endogenous retroviral-like particles, which if detected by innate immune sensors, could in turn induce the anti-viral IFNα cytokine, a hallmark of AGS and SLE (*Volkman and Stetson, 2014*). Here, we showed that the DNA hypomethylation observed in AGS samples affected LTR, LINE,

and to a lower extent, SINE repeated DNA elements (*Figure 4*). However, methylation pyrosequencing focused on human-specific LINE-1s showed that the extent of DNA hypomethylation at these elements, while significant, was minimal and that they in fact remained mostly methylated (*Figure 4—figure supplement 1B*). Likewise mining of our RNA-seq data sets or targeted real-time reverse-transcription PCR (RT-qPCR) assays aimed at the LINE-1 mRNA transcript failed to provide convincing evidence for LINE-1 transcriptional reactivation (data not shown). Finally, Western blots aimed at detecting the LINE-1 Orf1p protein were negative for AGS fibroblasts even though the protein was easily detected in Ntera2 control cells (data not shown). Altogether, while we can't rule out that LINE-1, or other retroelements, undergo reactivation in AGS cells, particularly in other cell types or at earlier time points during development, the evidence suggests that the DNA hypomethylation observed in AGS primary fibroblasts is not accompanied by LINE-1 reactivation.

The second major observation from this study is that all AGS subtypes tested here exhibit large accumulation of RNA:DNA hybrid species (*Figure 3*). RNA:DNA hybrids therefore represent candidate immunogenic nucleic acids in AGS. In contrast, ribonucleotide incorporation in genomic DNA was only observed in RNase H2-mutated samples (*Figure 2*). Our DRIP-seq profiling reveals that AGS2- and AGS4-specific RNA:DNA hybrids map to the intergenic space, away from traditional R-loop hotspots including genes, promoters, and terminators. AGS1- and AGS5-specific RNA:DNA hybrids were by contrast enriched over gene body regions (*Figure 3D*). All AGS-unique RNA:DNA hybrid peaks showed a lower overlap with GC skew, an important sequence determinant of R-loop formation (*Figure 3B*), suggesting that these hybrids may occur by a mechanism distinct from the well-accepted 'thread back' R-loop mechanism (*Aguilera and Garcia-Muse, 2012*). AGS-specific DRIP peaks were enriched in LINE-1 and LTR-containing sequences. This enrichment is intriguing in light of the proposed roles of SAMHD1, RNase H2, and TREX1 in the control of retroelement transposition (*Stetson et al., 2008*; *Zhao et al., 2013a*; *Volkman and Stetson, 2014*), although, as discussed above, we did not find convincing evidence for transposon reactivation here. Alternatively, it is possible that enrichment of AGS-specific RNA:DNA hybrids over LINE-1-rich regions simply reflects the fact that they map preferentially to intergenic and intronic regions. One possible clue as to the origin of AGS-specific DRIP peaks stems from the fact that a large fraction of such peaks in AGS2 and AGS4 samples overlap with AGS-specific PMDs in these two subtypes (*Figure 5*). Direct measurements in fact show that AGS-specific DRIP peaks are significantly hypomethylated in AGS1, AGS2, and AGS4 (*Figure 5*). Thus, the two main features observed in AGS samples, excessive RNA:DNA hybrid formation and DNA hypomethylation, often overlap, strongly supporting a link between both changes *in cis*. One possible explanation for this overlap is that regions experiencing DNA hypomethylation as a result of AGS mutations may undergo spurious transcription that lead to co-transcriptional R-loop formation (*Figure 7*). Another non-exclusive possibility is based on the proposition that PMD regions may result from DNA damage and inefficient maintenance DNA methylation (see above). It was recently described that RNA transcripts may serve as a template for the repair of double-strand DNA breaks upon hybridization to DNA via Rad52, a key member of the homologous recombination pathway. Interestingly, RNase H2 activity was shown to block the formation of these RNA:DNA hybrids (*Keskin et al., 2014*). It is therefore possible that some of the DNA breaks caused by AGS mutations and replication stress are repaired according to a pathway prone to RNA:DNA hybrid formation (*Figure 7*). While speculative, this model may also account for the lack of overlap with GC skew since those hybrids are thought to be mediated by Rad52 and not to arise co-transcriptionally. Experiments aimed at testing this proposal will be important.

One key unanswered question concerns the manner in which RNA:DNA hybrids trigger an immune response. The cGAS-STING pathway activates the interferon response upon sensing of not only cytosolic DNA (*Sun et al., 2013*), but also RNA:DNA hybrids (*Mankan et al., 2014*). Interestingly, several studies reported that multiple dsDNA break repair proteins can sense damaged DNA and induce a STING-mediated type I IFN response (*Zhang et al., 2011*; *Kondo et al., 2013*; *Hartlova et al., 2015*), thereby highlighting previously underappreciated links between the DNA damage response and innate immune signaling. Besides STING, it is worth noting that the endosomal receptor TLR9 has also been identified as an RNA:DNA hybrid sensor (*Rigby et al., 2014*) in addition to its canonical role in sensing unmethylated CpG-rich DNA (*Hemmi et al., 2000*). It is therefore possible that AGS mutations-induced DNA hypomethylation also contributes to the increased interferon response in AGS (*Figure 7*). Interestingly, treatment of colon adenocarcinoma with 5-aza-2'-deoxycytidine, which causes DNA damage and DNA demethylation, triggers a prominent interferon response (*Karpf et al., 1999*).

Future work will be necessary to dissect the pathways by which unmethylated DNA and RNA:DNA hybrids may be sensed to trigger the innate immune response characteristic of AGS, and by extension, SLE patients.

## Materials and methods

### Detection of incorporated ribonucleotides by nick translation assays

Detection of incorporated ribonucleotides was performed as previously described (*Hiller et al., 2012*). Briefly, 200 ng of human primary fibroblasts or yeast genomic DNA (wild-type and *RNASEH2*-deficient) was treated with 20 nM of purified recombinant human RNase H2 (*Loomis et al., 2014*) or water in RNase H2 reaction buffer (50 mM Tris-HCl pH 8, 60 mM KCl, 10 mM MgCl$_2$, 0.01% BSA (Bovine Serum Albumin), 0.01% Triton) at 37°C for 1 hr. 20 µM of unlabeled dATP, dGTP and dTTP, plus $3.7 \times 10^5$ Bq [α-$^{32}$P]-dCTP (PerkinElmer, Santa Clara, CA) and 5 U of *Escherichia coli* DNA polymerase I (New England Biolabs, Inc., Ipswich, MA) were added and the reaction was incubated at 16°C for 30 min and was run on a 1% TAE (Tris-acetate-EDTA) agarose gel. Visualization was performed using a Storm PhosphorImager, and bands were quantified using ImageQuant (GE Healthcare, United Kingdom). Relative ribonucleotide loads were calculated by dividing the radiolabel incorporation in the RNase H2-treated sample over the untreated sample. Experiments were performed at least in triplicate.

### DRIP-seq

DRIP-seq was performed on primary fibroblasts as previously described (*Ginno et al., 2012*). Sequencing reads were trimmed with FastqMcf (*Aronesty, 2011*) before mapping to the hg19 reference genome using BWA 0.6.1 (*Li and Durbin, 2009*). Peak calling was first performed by MACS 1.4.2 (*Zhang et al., 2008*) using input library as control. DRIP peaks were further assigned onto restriction fragments using BEDtools (*Quinlan and Hall, 2010*). For control and each AGS subtype, DRIP peaks from two independent samples were merged into one sample for downstream analysis. All overlap analysis was performed using BEDTools (*Quinlan and Hall, 2010*). GC skew annotation was according to low stringency SkewR peaks as previously described (*Ginno et al., 2013*). The enrichment or depletion of AGS-unique DRIP peaks over different genomic regions was measured as fold-change of percent base-pair overlap of AGS-unique DRIP peaks relative to common DRIP peaks. The statistical significance of enrichment or depletion of AGS-unique DRIP peaks over a specific genomic feature was measured as follows: for each AGS-unique peak, 500 shuffled peaks of equal lengths were extracted from the common peak set and their overlap with various genomic features was recorded as % length overlap. The significance of the difference in overlap between observed (AGS unique) and expected (shuffled) was calculated using empirical p-values according to the Monte Carlo method.

### Whole-genome bisulfite sequencing (MethylC-seq)

2 µg of genomic DNA was sheared by sonication down to 100–500 bp in size. Sequencing libraries were constructed as described before (*Schroeder et al., 2011*). We then performed bisulfite treatment using EZ DNA methylation-direct kit (Zymo Research, Irvine, CA) following manufacturer's instructions. 50 ng of bisulfite-treated DNA was amplified for 15 cycles using Pfu Cx Turbo Hotstart DNA polymerase (Agilent Technologies, Santa Clara, CA), and the quality of the library was checked on a 2100 Agilent Bioanalyzer prior to sequencing on an Illumina HiSeq 2000. Sequencing reads were trimmed as described in DRIP-seq. Mapping was performed using Bismark 0.7.7 (*Krueger and Andrews, 2011*), and percent methylation was called using custom Perl script, combind_strand_-meth.pl. C to T conversion rate was determined as the ratio of converted C in CHG and CHH context relative to the total number of CHG and CHH. CpG coverage was calculated as the average number of reads over CpG sites in the genome. Both C to T conversion rate and CpG coverage are reported in *Supplementary file 1*. To ensure good coverage and eliminate PCR bias, CpG sites with coverage below 4× or more than 99.9th percentile were discarded. Circos plot was generated using Circos 0.62–1 (*Krzywinski et al., 2009*). TSS and TTS metaplots were generated using custom Perl script, wig_to_metaplot_lowmem.pl. Methylation levels at different genomic regions were calculated BEDtools (*Quinlan and Hall, 2010*). Wilcoxon paired test was performed using R with the alternative hypothesis that the AGS samples are less methylated than the control. PMDs and HMDs were called

using StochHMM, a HMM-based software (*Lott and Korf, 2014*) as previously described (*Schroeder et al., 2013*) with modifications: PMDs were trained using random 25-kb regions with 25–55% methylation and HMDs were trained using random 25-kb regions with 60–100% methylation. H3K27me3 and H3K9me3 data sets were obtained from ENCODE normal human dermal fibroblasts (NHDF-Ad). The lamin B1 data set was obtained from Gene Expression Omnibus (GSE49341).

## Reduced representation bisulfite sequencing

Reduced representation bisulfite sequencing (RRBS) library was prepared according to the Myers lab protocol (*Varley et al., 2013*) prior to sequencing on an Illumina HiSeq 2000. Sequencing reads were trimmed with Trim Galore 0.3.3 (*Krueger, 2014*) and mapped to hg19 genome using Bismark 0.7.7 (*Krueger and Andrews, 2011*). TSS metaplot was generated using wig_to_metaplot_lowmem.pl.

## RNA-seq

RNA-seq libraries were constructed using the Illumina TruSeq kit prior to sequencing on an Illumina HiSeq 2000. Raw reads were trimmed as before and mapped to the hg19 genome using TopHat 2.0.5 (*Trapnell et al., 2009*). Read counts for each gene were calculated using HTSeq. Pearson's correlation between each pair of biological replicates was calculated by comparing every gene's log2 normalized read count (*Supplementary file 4*). Differential gene expression was identified using DESeq (*Anders and Huber, 2010*) using fold change > 2 and FDR (false discovery rate) < 0.1. Pathway analysis was performed using DAVID (*Huang da et al., 2009*).

## Data access

DRIP-seq, MethylC-seq, RRBS, and RNA-seq data are available at the Gene Expression Omnibus (GEO) database, under the accession number GSE57353. Custom Perl scripts are available at https://github.com/ywlim/Perl.

## RT-qPCR

Total RNA was extracted from 80% confluent primary fibroblasts using TRI reagent (Life Technologies, Grand Island, NY) and Direct-zol RNA miniprep kit (Zymo Research). RNA was reverse transcribed to first strand cDNA using iScript Reverse Transcription Supermix for RT-qPCR (Bio-Rad, Hercules, CA). cDNA was cleaned up using DNA clean and concentrator (Zymo Research) and resuspended in 10 μl water. For RT-qPCR, 5 μl of 1:50 dilution of the cDNA was used per well in a 20 μl reaction, along with 2 μl of 10 μM primer sets and 10 μl of SsoAdvanced Universal SYBR Green Supermix (Bio-Rad). Reactions were run in duplicate at least on a CFX96 Touch Real-Time PCR Detection System (Bio-Rad) with the following protocol: 95°C (2 min), 40 cycles of 95°C (10 s) and 60°C (15 s). Quantification was calculated using the CFX Manager software (Bio-Rad). Primer sequences are listed on *Supplementary file 3*.

## CRISPR/Cas knockout of RNASEH2A

RNASEH2A guide RNA sequence was designed using E-CRISP (http://www.e-crisp.org/) and the corresponding oligonucleotides (top: CACCGTTGGATACTGATTATGGCTC; bottom: AAACGAGCCATAATCAGTATCCAAC) and scramble oligonucleotides (top: CACCGGCACTAC CAGAGCTAACTCA; bottom: AAACTGAGTTAGCTCTGGTAGTGCC) were purchased from Life Technologies. The oligonucleotides were annealed and ligated into lentiCRISPR v2 (Addgene plasmid 52961) at the BsmBI restriction site. The resulting lentiCRISPR-RNASEH2A and lentiCRISPR-scramble plasmids were transformed into DH10B cells and purified using Qiagen Plasmid Midi Kit. To produce lentivirus carrying the CRISPR plasmids, 5 μg lentiCRISPR-RNASEH2A or lentiCRISPR-scramble, 4 μg psPAX2, and 1.5 μg of pMD2.G were transfected into HEK293T cells on a 10-cm plate using Turbofect (Life Technologies) following manufacturer's protocol. Viral supernatant was collected 48 hr later, filter sterilized with a 0.45-μm filter (Millipore, Billerica MA), and 1 ml of it was used to infect HeLa-GFP cells on a 6-well plate. 1 μg/ml puromycin was added 24 hr later. 22 days later, HeLa-GFP-RNASEH2A KO and HeLa-GFP-scramble cells were imaged using GFP microscopy. Whole-cell protein was extracted and Western blot was performed using antibodies against RNase H2A (1:1000, ProSci) or GFP (1:500, UC Davis/NIH NeuroMab Facility, Davis, CA), and tubulin (1:1000, Sigma–Aldrich, St. Louis, MO). Immunocytochemistry was performed on the same cells using antibody against γH2AX (1:150, Abcam, United Kingdom).

## LINE-1 bisulfite sequencing

600 ng of genomic DNA was bisulfite converted using EZ DNA methylation-direct kit (Zymo Research) and eluted in 30 μl elution buffer. 1 μl of it was then PCR amplified using two sets of LINE-1 primers (*Supplementary file 3*) (95°C for 4 min, 20 cycles of 95°C, 55°C and 72°C for 30 s each, 72°C for 4 min). PCR products were gel purified and cloned into pDRIVE cloning vector using Qiagen PCR cloning kit (Qiagen, Valencia, CA) and the resulting vectors were transformed into DH10B cells. Miniprep was performed using QIAprep Spin Miniprep Kit (Qiagen) and 16 clones were sequenced for each LINE-1 locus. Percent methylation at each CG site was the ratio of unconverted (CG) vs the sum of unconverted and converted (TG) molecules.

## Acknowledgements

We thank Dr Yanick Crow and Dr Gillian Rice for providing AGS patient samples, Dr Jodi Nunnari for the RNASEH2-deficient yeast cells, Dr Giovanni Capranico for the psPAX2 and pMD2.G plasmids, Dr David Segal for the HeLa cell line harboring silent GFP reporter gene, Dr John Moran for helpful discussions, and members of the Chedin lab for useful comments. This work was funded by grants from the Hartwell Foundation and the National Institutes of Health (NIH R01 GM094299) to FC. YWL was supported in part by a pre-doctoral training grant (5T32GM007377) and a Dissertation Year fellowship from UC Davis. This work used the Vincent J Coates Genomics Sequencing Laboratory at UC Berkeley, supported by NIH S10 Instrumentation Grants S10RR029668 and S10RR027303.

## Additional information

### Funding

| Funder | Grant reference | Author |
| --- | --- | --- |
| Hartwell Foundation | | Frédéric Chédin |
| National Institute of General Medical Sciences (NIGMS) | NIH R01 GM094299 | Frédéric Chédin |
| National Institute of General Medical Sciences (NIGMS) | 5T32GM007377 | Yoong Wearn Lim |

The funders had no role in study design, data collection and interpretation, or the decision to submit the work for publication.

### Author contributions

YWL, Conception and design, Acquisition of data, Analysis and interpretation of data, Drafting or revising the article; LAS, XX, Acquisition of data, Drafting or revising the article; SRH, Analysis and interpretation of data, Drafting or revising the article; FC, Conception and design, Analysis and interpretation of data, Drafting or revising the article

## Additional files

### Supplementary files

• Supplementary file 1. Genotype and high-throughput sequencing information for DRIP-seq, MethylC-seq, and RNA-seq.

• Supplementary file 2. Significantly up- and down-regulated genes in Aicardi–Goutières syndrome fibroblasts and their gene ontology analysis.

• Supplementary file 3. Primer sequences for real-time reverse-transcription PCR and bisulfite sequencing.

• Supplementary file 4. Pearson's correlation between each pair of RNA-seq biological replicates.

## Major dataset
The following dataset was generated:

| Author(s) | Year | Dataset title | Dataset ID and/or URL | Database, license, and accessibility information |
| --- | --- | --- | --- | --- |
| Lim YW | 2015 | DRIP-seq, RNA-seq and MethylC-seq datasets | http://www.ncbi.nlm.nih.gov/geo/query/acc.cgi?acc=GSE57353 | Publicly available at the NCBI Gene Expression Omnibus (Accession no: GSE57353) |

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
