## [Decision Letter]

Thank you for submitting your work entitled “Genome-wide DNA hypomethylation and RNA:DNA hybrid accumulation in Aicardi-Goutières syndrome” for peer review at *eLife*. Your submission has been favorably evaluated by James Manley (Senior editor), and three reviewers, one of whom is a member of our Board of Reviewing Editors. The following individuals responsible for the peer review of your submission have agreed to reveal their identity: Bing Ren (Reviewing editor); Paula Vertino (peer reviewer #3). A further reviewer remains anonymous.

The Reviewing editor has drafted this decision to help you prepare a revised submission.

Summary:

The work reports a number of interesting findings linking mutations of AGS1-5 to the symptoms displayed by a class of rare genetic disorders. Overall, the manuscript is well written, with most conclusions well supported by evidence presented. However, a number of over-interpretations or over-statements were made, where causal relationships were drawn based on insufficient data. The authors are encouraged to thoroughly revise the manuscript by toning down the statements or provide more objective statements.

Essential revisions:

1) Thoroughly revise the conclusions, where unwarranted causal relationship was drawn, as noted by both reviewers #2 and #3 (below). For example, “This study identifies global epigenetic perturbations and accumulation of RNA:DNA hybrids as two novel hallmarks that may drive the powerful immune response responsible for...”

2) Provide clearer description of the datasets. For example, provide information upfront regarding the WGBS sequence depth and resolution.

3) The RNAseq data only weakly support the idea of immune activation as a phenptype that is enriched in AGS fibroblasts. There were nearly as many downregulated genes...any patterns there? The data should be fully discussed.

4) The data in Figure 3 seem to suggest that the accumulation of RNA:DNA hybrids is confined to retroelements, and further, to specific classes (LINE,LTR) implying some specificity to the effect. Please discuss this trend and biological implications.

5) The Drip-seq data need to be better presented so that a reader can fully appreciate how the patterns change, and the spatial relationship to other genomic features (see Reviewer #3 comment below).

6) The DNA methylation data and RNA-DNA hybrid datasets also need to be presented more clearly, so that the spatial relationship between DNA methylation changes and accumulation of RNA:DNA hybrid can be better appreciated.

7) Experiments from the RNASEH2A k/o fibroblasts have revealed only a modest impact of RNASEH2A KO on DNA methylation at certain retroelements. Deeper analysis of this cell system would allow the investigator to determine the hierarchical nature of the genomic phenomena observed, and strength the causality statement.

8) Proper statistics needs to be used and documented when the enrichment/depletion characteristics and overlapping features are discussed.

Reviewer #1:

Aicardi-Goutières Syndrome (AGS) is caused by mutations in several genes involved in ribonucleotide metabolism and is characterized by an overt native immuneresponse, but the molecular causes of the symptoms remain unidentified. In this study, the authors profiled transcriptome, DNA methylome and RNA-DNA hybrids in fibroblasts from patients with AGS, and identified global DNA hypo-methylation and increased RNA:DNA hybrids as the most likely drivers of the inflammatory response in AGS patients. Importantly they highlight a potential epigenetic contribution to this syndrome (hypomethylation) specifically the reactivation of retrotransposon elements in the human genome, due to genome hypomethylation in the cells. The results are pretty compelling and the conclusions are well supported by the data provided.

1) Figure 1. It appears that for some genes, the normalized RNA-seq data for some of the samples (AGS4-P2, AGS5-P2, AGS2-P1) varies a lot between replicates in the same patient. Could you provide information showing the correlation between replicates?

2) Figure 3) This figure is confusing. It would help to explain how the right hand vertical axis corresponds to the left hand vertical axis. Figure 3 is much more easily interpretable and a cleaner presentation of the data.

3) The authors show that in AGS2 and AGS4 patients, regions containing RNA:DNA hybrids largely correspond with hypomethylated regions, but this is not true in AGS1 and AGS5 patients. Additionally, the authors mention that AGS1 and AGS5 patients have decrease hypomethylation. Patients AGS2 and AGS4 have RNASEH2a/2b mutations, and the authors show how RNASEH2a mutations trigger DNA hypomethylation. Is it possible to show the same for the RNASEH2b mutation corresponding with the AGS2 patient?

4) Figure 6) By focusing on only two LINE elements it seems too small of a set to “suggest a direct role of RNASEH2a in triggering DNA hypomethylation.” Perhaps you can suggest that it plays a role at these LINE elements. It is likely that if you focused on other areas you could see hypermethylation. Additionally in Figure 6 on cpg13, there are four reads supporting 50 % methylation level in the scramble, and two reads supporting a 0% methylation level in the KO. There are too few reads covering this cpg to make any claims regarding changes in methylation at cpg13.

Reviewer #2:

The study focuses on a potential mechanism connecting genetic mutations in TREX1, RNase H2 and SAMHD1 to inflammation-related problems in Aicardi-Goutières syndrome (AGS) in children. AGS involves the accumulation of incompletely metabolized nucleic acids resulting from the mutations. In this study, a variety of epigenome and transcriptome profiling, and a cutting-edge genetic engineering method are used to identify the source of immunogenic nucleic acids. The main conclusion is global epigenetic alterations and accumulation of RNA-DNA hybrids are involved in AGS, both shown here for the first time, and these alterations may drive immune responses that cause the inflammation.

Overall this is a novel study that provides a new evidence of two types of molecular changes that are variably associated with AGS, as shown in fibroblasts from AGS patients and controls.

The main issue I see with this manuscript is the overstatement of causality where only modest associations are shown. For example, I would agree with “This study identifies global epigenetic perturbations and accumulation of RNA:DNA hybrids as two novel hallmarks” but do not see adequate support for the remainder of the conclusion “that may drive the powerful immune response responsible for...” I also suggest adding additional caveat/caution about the first phrase as this study analyzed a very limited number of AGS samples, and they are heterogeneous in terms of the underlying mutation. Another example of overstatement is “our results indicate that AGS mutations in TREX1, RNASEH2A, RNASEH2B, and SAMHD1 lead to the accumulation of RNA:DNA hybrids over retrotransposon-rich intergenic regions.” The data do not support “lead to”, but perhaps better described as “associated with”.

The WGBS data is quite interesting and the Circos display allows a reasonable visual comparison and highlights global differences. I have minor concerns about the data. The coverage is quite low, 3x to 7x, which defeats the main benefit of this method, base resolution. While the overall conclusion is probably correct, it would be better for the reader to know this is very low pass sequencing upfront, rather than putting it in a supplemental table only. It is also not clear how the C to T conversion was calculated. I could not find an explanation in the table or methods.

Reviewer #3:

The manuscript by Lim et al. makes some very interesting observations regarding the molecular basis of Aicardi-Goutieres syndrome, an autoimmune disorder of uncertain molecular origin despite well described loss of function mutations in several nucleic acid processing enzymes. The authors find that loss of function of AGS genes is associated, to varying degrees, with widespread hypomethylation of DNA as well as the selective accumulation of RNA:DNA hybrids in intergenic regions and over repetitive elements, and modest changes in gene expression. How these three observations are molecularly linked is less clear. The manuscript is well written and the experiments appropriately controlled. For the most part (exceptions noted below, the conclusions drawn are supported by the data provided.

1) The authors appear to favor the idea that RNA:DNA hybrid accumulation is the culprit behind activation of the immune response in AGS cells, though the data at hand are such that a direct causal link cannot be established. How this might be achieved is unclear, but the Discussion proposes a number of ‘sensing’ mechanisms, some of which would require a mechanism in cis (e.g. detection of hybrids as they are formed) whereas others (TLR, GAS-STING etc. ) require cytoplasmic accumulation of duplexes. Is there any evidence in the AGS fibroblasts or in the RNASeH2 K/O fibroblasts of non-genomic accumulation of DNA:RNA duplexes in AGS cells?

2) The RNAseq data only weakly support the idea of immune activation as a phenptype that is enriched in AGS fibroblasts. There were nearly as many downregulated genes...any patterns there? The data should be fully discussed.

3) The DNA damage known to accompany defects in the AGS genes is proposed as intermediates that could mediate and/or cause the RNA:DNA hybrids or DNA hypomethylation. Do RNA:DNA hybrids accumulate at the sites of ectopic DNA damage ? This is something that could be addressed in the RNAseH2 k/o cells which exhibit by co-localization studies with gamma H2Ax.

4) It is suggested that the relationship of RNA:DNA hybrid accumulation to retroelements might be circumstantial, due to the enrichment of these structures in intragenic regions... however, the data in Figure 3, if representative, seem to suggest the opposite; that the accumulation of RNA:DNA hybrids is confined to retroelements, and further, to specific classes (LINE,LTR) implying some specificity to the effect. Are the unique drip seq signals by and large confined to retroelements as shown in Figure 3? Is it possible that you are detecting some intermediate in piRNA-mediated transposon silencing?

5) The Drip-seq data are presented in a very general way (as total # of peaks, distance covered etc.) making it difficult to fully appreciate how the patterns change, and the spatial relationship to other genomic features. Both and increase in burden and size is implied. Statements like “...Drip-seq peaks were 3-4 fold larger in size” are unclear. Do you mean that existing Drip-seq peaks in control cells spread / expand laterally in AGS cells, or is the trend for new peaks to form in patient cells? Figure 3 – the ‘raw’ data on common and unique sites would be more informative rather than “fold change” e.g. what fraction of common and unique peaks overlap each genomic feature? This would provide not only a comparison between unique and common sites, but also an indication of the relative distribution of Dripseq peaks across genomic features in each setting.

6) Likewise, it is hard to appreciate the spatial relationship between DNA methylation changes and accumulation of RNA:DNA hybrids, given that the former are very broad and might encompass multiple retroelements some of which change and others that don't. If one looks specifically at those AGS specific DRIP-seq peaks, what fraction are overlapping retroelements, and what is the average DNA methylation level of these in control and AGS cells?

7) The lack of overlap between AGS unique DRIP seq peaks and regions of GC skew/ annotated TSSs is taken as an indication that the aberrant RNA:DNA hybrids detected do not arise co-transcriptionally. Barring a direct test of the role of transcription on RNA: DNA hybrids in AGS cells, I'm not sure one can rule out a transcription-dependent mechanism at this juncture.

8) The authors interpret the data from the RNASEH2A k/o fibroblasts as evidence that “...RNAse H2 defects directly drive these epigenetic perturbations”. Thus far they have only observed a modest impact on DNA methylation at certain retroelements. Whether there is a measurable global (bulk) effect on DNA methylation as in AGS cells is not clear, nor has the impact on RNA:DNA hybrid formation been studied. This could be a powerful system as it might allow the investigator to determine the hierarchical nature of the genomic phenomena observed.

9) Why/what are the missing bubbles in the bisulfite sequencing data in Figure 6? Depending on the nature of these CpGs (unmethylated/methylated) would change the magnitude of the methylation change significantly.

---

## [Author Response]

*1) Thoroughly revise the conclusions, where unwarranted causal relationship was drawn, as noted by both reviewers #2 and #3 (below)*. *For example, “This study identifies global epigenetic perturbations and accumulation of RNA:DNA hybrids as two novel hallmarks that may drive the powerful immune response responsible for...”*

We systematically went through the manuscript and eliminated or qualified any unwarranted causal relationship. For example, the sentence above (which appeared in the impact statement) was modified to “Global epigenetic perturbations and accumulation of RNA:DNA hybrids were identified as two novel hallmarks of the lupus-like inflammatory disorder Aicardi-Goutières syndrome”. As suggested by Reviewer #2, we inserted a clause at the top of the Discussion stating that “additional samples encompassing other AGS mutations subtypes will need to be studied to confirm this trend”. Likewise, instances of “lead to” were changed to “associated with”. As suggested by Reviewer #3, we inserted a clause indicating at the end of the Results section that the effect of RNASEH2A deficiency in triggering DNA hypomethylation was observed “at least for the loci studied here”.

*2) Provide clearer description of the datasets. For example, provide information upfront regarding the WGBS sequence depth and resolution*.

We now indicate coverage depth in the Results section and added an explanation of the method used to calculate bisulfite conversion efficiency in Materials and methods.

3) The RNAseq data only weakly support the idea of immune activation as a phenptype that is enriched in AGS fibroblasts. There were nearly as many downregulated genes...any patterns there? The data should be fully discussed.

We now include a gene ontology analysis for both down- and up-regulated genes. This was added to the Results section and to [Supplementary-material SD2-data]. The data indicates that AGS down-regulated genes significantly associate with ontologies linked to cell adhesion and the extracellular matrix ([Supplementary-material SD2-data]). These ontologies are interesting given the fact that skin lesions are a clinical hallmark of AGS and SLE patients.

*4) The data in*
Figure 3
*seem to suggest that the accumulation of RNA:DNA hybrids is confined to retroelements, and further, to specific classes (LINE,LTR) implying some specificity to the effect. Please discuss this trend and biological implications*.

We apologize if the genome browser screenshots gave the impression that accumulation of RNA:DNA hybrids was confined to retroelements. This is certainly not our view. To address this, we added Figure 3 as an example of AGS-unique DRIP peaks overlapping with an intergenic non-retroelement region. We also significantly expanded the corresponding paragraph in the Results section to further discuss the distribution of AGS-specific DRIP peaks over repeat elements. We now specifically state that “whether the reported enrichment of AGS-specific DRIP peaks over LINE and LTR repeats carries biological significance or is simply a reflection of the increased repeat content of intergenic and intronic space remains to be determined”. As was mentioned in detail in the Discussion, we did not find convincing evidence of retroelement reactivation in AGS fibroblasts and stated that “it is possible that enrichment of AGS-specific RNA:DNA hybrids over LINE-1-rich regions simply reflects the fact that they map preferentially to intergenic regions”. Nevertheless, as was stated in the Discussion, we agree that the observation of RNA:DNA hybrids over retroelements is intriguing “in light of the proposed roles of SAMHD1, RNase H2, and TREX1 in the control of retroelement transposition”. In the spirit of open-mindedness, our proposed model (Figure 7) therefore included “defective retroelement control” as a potential source of RNA:DNA hybrid accumulation.

*5) The Drip-seq data need to be better presented so that a reader can fully appreciate how the patterns change, and the spatial relationship to other genomic features (see Reviewer #3 comment below)*.

We agree with this and have significantly expanded the corresponding Results section. We now provide a much more detailed description of the distribution of R-loops commonly formed in both control and patient samples over genomic features including promoters, gene bodies, terminal regions and intergenic space. These additions can be found in the subsection “RNA:DNA hybrids accumulate in AGS patients” of the manuscript and in the new Figure 3—figure supplement 1. As requested by Reviewer #3, we also added Figure 3—figure supplement 1 to show the fraction of common and unique DRIP peaks overlapping with each genomic feature. We believe that this more thorough representation of the data strengthened the manuscript and thank the reviewers for the suggestion.

*6) The DNA methylation data and RNA-DNA hybrid datasets also need to be presented more clearly, so that the spatial relationship between DNA methylation changes and accumulation of RNA:DNA hybrid can be better appreciated*.

We agree that properly describing the spatial relationship between DNA methylation and the accumulation of RNA:DNA hybrids is essential. In the original version, the relationship between RNA:DNA hybrid and partially methylated domains (PMDs) was reported in Figure 5, which showed that a significant portion of AGS2 and AGS4-unique RNA:DNA hybrid peaks fell onto AGS2 and AGS4-unique PMDs. To expand this data and facilitate its comprehension, we added Figure 5, a genome browser screenshot that clearly shows the association between peaks of RNA:DNA hybrids in AGS2 and AGS4 subtypes with the presence of a partially methylated region in these two samples. We also added Figure 5 to more quantitatively address the relationship between the presence of a peak of RNA:DNA hybrid and the DNA methylation levels of that peak in control versus patient samples. As now shown, the formation of an extra peak of RNA:DNA hybrid in AGS1, AGS2, and AGS4 (but not AGS5) is associated with a significant decrease in the DNA methylation status of that region in patient versus control. This new data solidifies the overlap between changes in RNA:DNA hybrid formation and DNA hypomethylation *in cis*, a point that has also been added in the Discussion.

*7) Experiments from the RNASEH2A k/o fibroblasts have revealed only a modest impact of RNASEH2A KO on DNA methylation at certain retroelements. Deeper analysis of this cell system would allow the investigator to determine the hierarchical nature of the genomic phenomena observed, and strength the causality statement*.

While we fully agree that further experiments using the RNASEH2A KO system will be informative, we believe that our findings are sufficiently supported by the fact that knockout of the RNASEH2A gene was sufficient to cause partial loss of DNA methylation at dispersed LINE-1 loci and the reactivation of a previously silent reporter gene. We intend to use the knockout system to further investigate the mechanisms linking RNase H2 function, DNA replication stress, RNA:DNA hybrid formation and DNA hypomethylation, but these studies will be reported in a different manuscript.

*8) Proper statistics needs to be used and documented when the enrichment/depletion characteristics and overlapping features are discussed*.

We have now included statistical tests in Figure 3 and Figure 3—figure supplement 1. Likewise, we added statistics to Figure 5 and to the new panel in Figure 5.

Reviewer #1:

*1)*
Figure 1*. It appears that for some genes, the normalized RNA-seq data for some of the samples (AGS4-P2, AGS5-P2, AGS2-P1) varies a lot between replicates in the same patient. Could you provide information showing the correlation between replicates?*

We have added [Supplementary-material SD4-data], which provides Pearson’s correlation between each pair of RNA-seq replicates.

*2)*
Figure 3*) This figure is confusing. It would help to explain how the right hand vertical axis corresponds to the left hand vertical axis.*
Figure 3
*is much more easily interpretable and a cleaner presentation of the data*.

We apologize for the confusion. Figure 3 is meant to convey the extent of overlap of DRIP peaks between the different samples, similar to a Venn diagram. The right hand label was not an axis and was added to assist the understanding of the graph – we have clarified this in the figure legend. See also response to Essential revision #5.

*3) The authors show that in AGS2 and AGS4 patients, regions containing RNA:DNA hybrids largely correspond with hypomethylated regions, but this is not true in AGS1 and AGS5 patients. Additionally, the authors mention that AGS1 and AGS5 patients have decrease hypomethylation*. *Patients AGS2 and AGS4 have RNASEH2a/2b mutations, and the authors show how RNASEH2a mutations trigger DNA hypomethylation. Is it possible to show the same for the RNASEH2b mutation corresponding with the AGS2 patient?*

We agree that it will be interesting to show that *RNASEH2B* defect triggers DNA hypomethylation. However, due to time constraints we have not yet generated a knockout system for *RNASEH2B*. The RNase H2 complex is composed of 3 subunits: RNase H2A, 2B and 2C, with 2A being the catalytic subunit. Therefore, we focused our effort on characterizing the *RNASEH2A* gene.

*4)*
Figure 6*) By focusing on only two LINE elements it seems too small of a set to “suggest a direct role of RNASEH2a in triggering DNA hypomethylation.” Perhaps you can suggest that it plays a role at these LINE elements. It is likely that if you focused on other areas you could see hypermethylation. Additionally in*
Figure 6
*on cpg13, there are four reads supporting 50 % methylation level in the scramble, and two reads supporting a 0% methylation level in the KO. There are too few reads covering this cpg to make any claims regarding changes in methylation at cpg13*.

We agree that hypomethylation at LINE elements may not represent loss of methylation at other loci. We have adjusted the text to indicate that the loss of methylation was observed “at least for the loci studied here” or “at a subset of LINE-1 loci” to clarify our claims. Our statistical test in Figure 6 was performed by comparing all 13, instead of individual, CpG sites between scramble and the KO. Therefore, while we cannot make any claims regarding changes in methylation at CpG site #13 specifically, we can say the surveyed loci in LINE-1 as a whole are hypomethylated in the KO.

Reviewer #2:

*The main issue I see with this manuscript is the overstatement of causality where only modest associations are shown. For example, I would agree with “This study identifies global epigenetic perturbations and accumulation of RNA:DNA hybrids as two novel hallmarks” but do not see adequate support for the remainder of the conclusion “that may drive the powerful immune response responsible for...” I also suggest adding additional caveat/caution about the first phrase as this study analyzed a very limited number of AGS samples, and they are heterogeneous in terms of the underlying mutation. Another example of overstatement is “our results indicate that AGS mutations in TREX1, RNASEH2A, RNASEH2B, and SAMHD1 lead to the accumulation of RNA:DNA hybrids over retrotransposon-rich intergenic regions.” The data do not support “lead to”, but perhaps better described as “associated with”*.

*The WGBS data is quite interesting and the Circos display allows a reasonable visual comparison and highlights global differences. I have minor concerns about the data. The coverage is quite low, 3x to 7x, which defeats the main benefit of this method, base resolution. While the overall conclusion is probably correct, it would be better for the reader to know this is very low pass sequencing upfront, rather than putting it in a supplemental table only. It is also not clear how the C to T conversion was calculated. I could not find an explanation in the table or methods*.

We appreciate the reviewer’s comments and suggestions. Please see our replies to Essential Revisions #1 and #2.

Reviewer #3:

*1) The authors appear to favor the idea that RNA:DNA hybrid accumulation is the culprit behind activation of the immune response in AGS cells, though the data at hand are such that a direct causal link cannot be established. How this might be achieved is unclear, but the Discussion proposes a number of ‘sensing’ mechanisms, some of which would require a mechanism in cis (e.g. detection of hybrids as they are formed) whereas others (TLR, GAS-STING etc. ) require cytoplasmic accumulation of duplexes*. *Is there any evidence in the AGS fibroblasts or in the RNASeH2 K/O fibroblasts of non-genomic accumulation of DNA:RNA duplexes in AGS cells?*

We agree that it may be an over interpretation to attribute the immune response solely to RNA:DNA hybrid accumulation in AGS cells. We have therefore eliminated or qualified any unwarranted causal relationship (please see our reply to Essential Revision #1). We also agree that while numerous studies show sensing of nucleic acids either in the cytoplasm or as a consequence of DNA damage, the mechanisms underlying such sensing and their distribution within cellular sub-compartments remain unclear. We have thus far not specifically looked at non-genomic RNA:DNA hybrid accumulation but plan onto doing this in the near future.

*2) The RNAseq data only weakly support the idea of immune activation as a phenptype that is enriched in AGS fibroblasts. There were nearly as many downregulated genes...any patterns there? The data should be fully discussed*.

We added gene ontology analysis for both up- and down-regulated genes in [Supplementary-material SD2-data]. A short description of these ontologies for down-regulated genes was added to the Results section. See response to Essential Revision #3 for further information.

*3) The DNA damage known to accompany defects in the AGS genes is proposed as intermediates that could mediate and/or cause the RNA:DNA hybrids or DNA hypomethylation. Do RNA:DNA hybrids accumulate at the sites of ectopic DNA damage ? This is something that could be addressed in the RNAseH2 k/o cells which exhibit by co-localization studies with gamma H2Ax*.

This is undoubtedly an interesting area of future investigation. At this time, however, we do not have data directly relevant to this point and believe it is beyond the scope of this manuscript.

*4) It is suggested that the relationship of RNA:DNA hybrid accumulation to retroelements might be circumstantial, due to the enrichment of these structures in intragenic regions... however, the data in*
Figure 3*, if representative, seem to suggest the opposite; that the accumulation of RNA:DNA hybrids is confined to retroelements, and further, to specific classes (LINE,LTR) implying some specificity to the effect. Are the unique drip seq signals by and large confined to retroelements as shown in*
Figure 3*? Is it possible that you are detecting some intermediate in piRNA-mediated transposon silencing?*

Please see response to Essential Revision #4.

*5) The Drip-seq data are presented in a very general way (as total # of peaks, distance covered etc.) making it difficult to fully appreciate how the patterns change, and the spatial relationship to other genomic features. Both and increase in burden and size is implied. Statements like “...Drip-seq peaks were 3-4 fold larger in size” are unclear. Do you mean that existing Drip-seq peaks in control cells spread / expand laterally in AGS cells, or is the trend for new peaks to form in patient cells?*
Figure 3
*– the ‘raw’ data on common and unique sites would be more informative rather than “fold change” e.g. what fraction of common and unique peaks overlap each genomic feature? This would provide not only a comparison between unique and common sites, but also an indication of the relative distribution of Dripseq peaks across genomic features in each setting*.

We agree with this point – please see answer to Essential Revision #5.

6) Likewise, it is hard to appreciate the spatial relationship between DNA methylation changes and accumulation of RNA:DNA hybrids, given that the former are very broad and might encompass multiple retroelements some of which change and others that don't. If one looks specifically at those AGS specific DRIP-seq peaks, what fraction are overlapping retroelements, and what is the average DNA methylation level of these in control and AGS cells?

Please see answer to Essential Revision #6. Figure 5 now describes the methylation state of AGS-unique DRIP peaks clearly showing DNA hypomethylation in AGS1, AGS2, AGS4, but not AGS5 samples.

In answer to your specific request regarding the methylation state of AGS-unique DRIP peaks that overlap retroelements, we are providing Figure 8.

Author response image 1.**DOI:**
http://dx.doi.org/10.7554/eLife.08007.020

*7) The lack of overlap between AGS unique DRIP seq peaks and regions of GC skew/ annotated TSSs is taken as an indication that the aberrant RNA:DNA hybrids detected do not arise co-transcriptionally. Barring a direct test of the role of transcription on RNA: DNA hybrids in AGS cells, I'm not sure one can rule out a transcription-dependent mechanism at this juncture*.

We agree and have edited the language throughout the manuscript so as not to rule out possible co-transcriptional formation of AGS-unique peaks of RNA:DNA hybrids.

*8) The authors interpret the data from the RNASEH2A k/o fibroblasts as evidence that “...RNAse H2 defects directly drive these epigenetic perturbations”. Thus far they have only observed a modest impact on DNA methylation at certain retroelements. Whether there is a measurable global (bulk) effect on DNA methylation as in AGS cells is not clear, nor has the impact on RNA:DNA hybrid formation been studied. This could be a powerful system as it might allow the investigator to determine the hierarchical nature of the genomic phenomena observed*.

Please see our reply to Essential Revision #7.

*9) Why/what are the missing bubbles in the bisulfite sequencing data in*
Figure 6*? Depending on the nature of these CpGs (unmethylated/methylated) would change the magnitude of the methylation change significantly*.

The missing bubbles correspond to CpG sites that are absent from the sequenced molecule. This is likely due to heterogeneity in LINE-1 sequences. We added a sentence clarifying this in the figure legend.